# Electrocardiographic and Seasonal Patterns Allow Accurate Differentiation of Tako-Tsubo Cardiomyopathy from Acute Anterior Myocardial Infarction: Results of a Multicenter Study and Systematic Overview of Available Studies

**DOI:** 10.3390/biom9020051

**Published:** 2019-01-30

**Authors:** Seher Çatalkaya Demir, Erdem Demir, Sibel Çatalkaya

**Affiliations:** 1Department of Cardiology, Uşak State Hospital, 64200 Uşak, Turkey; seher.catalkaya@web.de (S.Ç.D.); e.demir1@saglik.gov.tr (E.D.); 2Department of Cardiology, University Clinic Uşak, 64200 Uşak, Turkey

**Keywords:** Tako-Tsubo cardiomyopathy, STEMI, ECG, seasonal variation, chronobiology

## Abstract

**Background**. Though several studies about prevalence, etiology, clinical characteristics, preceding events, clinical management, and outcome of Tako-Tsubo cardiomyopathy (TTC) exist, the current knowledge of TTC remains limited. **Objective**. In 2006, TTC was classified among the acquired forms of cardiomyopathy. On the basis of pathophysiological implications, we analyzed whether the presence of ST-segment elevation in lead -aVR (i.e., ST-segment depression in aVR) and the simultaneous absence of ST-segment elevation in lead V1 allow a reliable differentiation of TTC from acute anterior ST-segment elevation myocardial infarction (STEMI). A further investigative feature is the seasonal variation of TTC. Since acute cardiovascular events exhibit definite chronobiological patterns, various small studies have tried to evaluate whether this is also the case for TTC. Because results are conflicting, we also conducted a multicenter study and analyzed the findings in context with a systematic overview of available studies. **Methods**. We compared the ECG patterns of 115 patients with TTC, who were admitted to five large acute cardiac care centers associated with university hospitals in Southwestern Germany between January 2001 and June 2011, with those of 100 patients with acute anterior ST-segment elevation myocardial infarction (STEMI) treated in one of these centers. In addition, we performed a computer-assisted MEDLINE search of the literature from January 2000 to September 2011 and analyzed the chronobiological patterns of available TTC cases, including our TTC cohort. **Results**. Testing the predefined diagnostic criteria was superior to any other electrocardiographic finding and differentiated TTC from anterior STEMI with a sensitivity of 73%, a specificity of 84%, a positive predictive value of 63%, and a negative predictive value of 89%. Beyond that, the onset of TTC showed a clear variation as a function of season and month. While events occurred most frequently during summer (38.4%, *p* < 0.01), the event rate was the lowest in autumn (16.4%) and winter (21.9%). Chronobiological analyses on a monthly basis identified a significant annual rhythmic pattern in TTC, which peaked in August (11.9%; *p* < 0.01) and had its nadir in November (6.3%). **Conclusions**. Our data illustrate that the ST-segment changes in leads aVR and V1 represent a simple and accurate ECG criterion to differentiate TTC from anterior STEMI in patients who are admitted within 12 h of symptom onset. Similarly, the results of our seasonal analysis indicate a distinct chronobiological variation in TTC occurrence. TTC, thereby, differs from major acute cardiovascular diseases, especially acute myocardial infarction (AMI), which is characterized by winter peaks and troughs in summer. If these results are confirmed in large independent cohorts, they may yield diagnostic implications, changing the regular invasive AMI management in TTC patients.

## 1. Introduction

Tako-Tsubo cardiomyopathy (TTC), also known as stress cardiomyopathy, was first described in 1990 in Japan [1] and derives its name from the Japanese word tako-tsubo, which means ’octopus trap’ and is referred to the typical end-systolic apical ballooning of the left ventricle in the acute phase of its presentation. In 2006, TTC was classified among the acquired forms of cardiomyopathy in the contemporary definition and according to the classification of cardiomyopathies proposed by the American Heart Association [2]. TTC is a non-ischemic cardiomyopathy in the absence of relevant atherosclerotic coronary artery disease (CAD), which is characterized by acute but often rapidly reversible left ventricular systolic dysfunction with regional wall-motion abnormalities that extend beyond a single coronary vascular bed and is often associated with a precipitating stressor. This distinct form of ventricular stunning typically affects older women and preferentially involves the distal portion of the left ventricle (LV) (‘apical ballooning’), which is compensated by hypercontractility of the basal inferior and anterior LV. It is estimated that TTC is found in about 2% of patients presenting with acute coronary syndrome (ACS). The diagnosis of TTC has important implications for its clinical management. Although its presentation often mimics ST-segment elevation myocardial infarction (STEMI) of the anterior wall, the outcome seems favorable with appropriate symptomatic therapy. Nevertheless, the rate of adverse events should not be underestimated. Even more, it is of substantial interest to investigate differentiation criteria to distinguish TTC from other acute cardiac incidents. One small study suggested a discrete electrocardiographic pattern in Asian TTC patients [3]. On the basis of pathophysiological implications, we investigated whether the presence of ST-segment elevation in lead -aVR (i.e., ST-segment depression in aVR) and the simultaneous absence of ST-elevation in lead V1 allows reliable differentiation from acute anterior STEMI. Besides that, it is well known that acute cardiovascular events exhibit definite chronobiological patterns. Various small studies have tried to evaluate whether this is also the case for TTC. Because results are conflicting, we conducted a multicenter study and analyzed the findings in context with a systematic overview of available studies [4,5,6,7,8,9,10]. The data were acquired from Central, Western, and Southern Europe, as well as from the United States. Consequently, different climatic and geographic circumstances are reflected in our results.

## 2. Materials and Methods

### 2.1. Study Population

#### 2.1.1. Electrocardiographic Pattern

To accomplish an optimal evaluation of the electrocardiographic segments, patients with bundle-branch block, left-ventricular hypertrophy, or atrial fibrillation and coronary artery disease requiring a percutaneous intervention were excluded. We performed the final analysis in a sub-cohort of 106 TTC and 85 STEMI patients, who were admitted within 12 h of symptom onset to five large acute cardiac care centers associated with university hospitals in Southwestern Germany between 2001 and 2011 for TTC and between 2009 and 2010 for anterior STEMI. The study was approved by the local ethics committees (ethical code 197/10-UBB/bal.). All patients fulfilled the Mayo Clinic diagnosis criteria for TTC, which in particular include: (a) a transient left ventricular wall motion abnormality preferred in the midventricular and/or apical segments, which typically does not fit to a single coronary artery perfusion territory, (b) the absence of obstructive coronary artery disease or acute plaque rupture in the coronary angiography, though a coronary artery disease can coincidentally be presented, (c) new electrocardiographic changes, (d) at most, moderate elevation of cardiac ischemia markers, (e) a potential stressor (e.g., emotional, physical), (f) exclusion of pheochromocytoma and myocarditis. All patients underwent coronary angiography to rule out CAD. The clinical variables were recorded in part prospectively (from July 2009). Twelve lead ECGs were recorded at a paper speed of 25 mm/s and amplification of 10 mm/mV. Deviations of the ST-segment from the baseline PR-segment were measured 80 ms after the J point and were considered significant when exceeding 0.5 mm in limb and 1.0 mm in precordial leads [11]. To ensure a general standardization, we used the Cabrera format in an anatomically contiguous manner from left superior-basal to right inferior to display the limb leads: aVL, I, -aVR (i.e., lead aVR with reversed polarity), II, aVF, and III in 30° angular distance. We evaluated sensitivity and specificity, as well as the positive and negative predictive values for ST-segment elevation in -aVR (i.e., ST-segment depression in aVR) and the simultaneous absence of ST-segment elevation in V1.

#### 2.1.2. Seasonal Variation

To assess the seasonal pattern of TTC, we analyzed the chronobiological occurrence of our TTC cohort. The study was approved by the local ethics committees, and informed consent was obtained from all patients. Clinical variables were recorded retrospectively (from January 2001 to June 2009) and in part prospectively (from July 2009 to June 2011) on a standardized form. Retrospective data were actively searched in hospital discharge records and coronary care units as well as in cardiac catherization laboratory databases. The day of onset of symptoms was categorized both into one-month intervals and, for the analysis of seasonal variation, into three-month intervals divided in accordance to the more traditional astronomical seasonal definition, which most likely considers astrophysiological conditions and circumstances depending on earth’s rotation (spring: March 21 to June 20; summer: June 21 to September 22; autumn: September 23 to December 20; and winter: December 21 to March 20). Because of our limited cohort number, we additionally performed a computer-assisted MEDLINE search of the literature (from January 2000 to September 2011) using the following search terms: TTC, transient left ventricular apical ballooning syndrome, tako-tsubo-like left ventricular dysfunction, ampulla cardiomyopathy, tako-tsubo, and apical ballooning. The criteria for inclusion in our analyses were set a priori and included: (a) reporting of original data, (b) for the purpose of representativeness, inclusion of at least 30 cases, and (c) adherence to the requested diagnostic criteria for TTC. In the case of multiple publications describing an intersection of cases, we used the latest ones. The heterogeneity of the methodologies used among studies did not allow a formal meta-analysis; thus, data are presented in a narrative synthesis.

### 2.2. Statistical Analysis

Statistical analyses were performed using SPSS Statistics version 14 and Microsoft Ecxel XP for Windows. Statistical significance was defined as two-sided *p* < 0.05. The study population was described in accordance with sociodemographic and clinical characteristics. Continuous variables are given as mean ± SD or medians (IQR). Categorial variables are expressed as percentages and numbers. Comparisons of continuous variables were analyzed by the t-test. The distribution of symptom onset within the three-month periods was tested for uniformity in the overall population by the chi-square test. Moreover, a chronobiological analysis of the seasonal patterns of TTC was performed by applying a partial Fourier analysis to the monthly data using validated Chronolab software.

## 3. Results

### 3.1. Demographic and Clinical Features

The mean age of our TTC study population at the index event was 66.6 ± 12.7 years, and the majority of patients were women (n = 101; 87.8%; *p* < 0.01), whereas the mean age of our anterior STEMI cohort was 64.3 ± 14.1 years with a majority of male patients (n = 73). The main complaints at admission did not differ from those of patients with acute myocardial infarction (see Table 1). In several cases, patients with headache and backache (n = 8), general weakness (n = 7), epigastric pain (n = 4), apoplexy with hemiparesis (n = 1), visual field defects (n = 1), tingling paresthesia (n = 1), urinary retention (n = 1), and pre-collapse (n = 1) were recorded. None of the patients presented a relevant pre-existing cardiovascular disease (e.g., obstructive coronary artery disease, past myocardial infarction, or previous revascularization). In two cases (1.7%), coronary artery plaque was recognized. A reduced left ventricular ejection fraction was noted in 72 (62.6%) of our TTC patients. The assessment of the systolic left ventricular ejection fraction (EF) was performed either echocardiographically, using Simpson’s or Teichholz’s formula, or on the basis of data collected during coronary angiography, if available. A recovery of the systolic left ventricular function was registered for 38 patients before discharge. None of our TTC patients showed a hemodynamic relevant valvular disease. In two cases (1.7%), an apical left ventricular thrombus with a diameter of 1.9 × 0.9 cm and 1.3 × 0.7cm was obtained, requiring a mini-thoracotomy in one case. The data regarding non-invasive and invasive clinical conditions, including in-hospital and long-term outcomes, are shown in Table 1. Considering the main purpose of this research, we did not observe detailed clinical features of anterior STEMI patients.

### 3.2. Triggering Life Events

TTC is, in most cases, accompanied by emotional, physical, and clinical stressors. Cases without any associated stimulus can also occur. Table 2 shows an overview of known triggering factors. Eight of our TTC patients (7%) had no preceding event on symptom onset.

### 3.3. Laboratory Values

Since varying biomarkers, e.g., troponin I/T/, brain natriuretic peptide/N-terminal pro-brain natriuretic peptide (BNP/NT-proBNP) and test types, e.g., high-sensitive versus conventional test were used at different times in different centers, the indicated means and medians were only partially comparable (Table 3). While, in our TTC cohort, biomarkers on admission showed at most a moderate increase, the on-hospitalization cardiac enzymes and hsCRP (high-sensitivity C-reactive protein in our anterior STEMI patients were markedly increased (Table 4). We did not perform a laboratory enzyme course observation for our STEMI patients.

### 3.4. Electrocardiographic Pattern

In Figure 1, representative ECGs of TTC and anterior STEMI patients are shown. In initial analyzes, predefined electrocardiographic criteria exhibited a clear trend allowing differential diagnosis of TTC (data not shown). After exclusion of those subjects who were admitted to hospital later than 12 h after symptom onset, a statistically significant distinct pattern of ST-segment elevations was observed. As seen in Figure 2a, TTC was more frequently associated with ST-segment elevation in the limb leads aVF/II/I and especially in lead -aVR (all *p* < 0.01), while STEMI of the anterior wall (Figure 2b) showed more frequently ST-segment elevations in V2/V3/V4 and especially lead V1 (all *p* < 0.01). When testing the predefined diagnostic criteria, we found that the combined presence of ST-segment elevation in lead -aVR (e.g., ST-segment depression in aVR) and the absence of ST-segment elevation in lead V1 were superior to any other electrocardiographic finding and differentiated TTC from STEMI with a sensitivity of 73%, a specificity of 84%, a positive predictive value of 63%, and a negative predictive value of 89% (Table 5).

### 3.5. Seasonal Variation

We identified seven studies with data on seasonal and monthly variations, which reported a total of 537 cases including our TTC patients (n = 115). In this total population, the onset of TTC showed a clear variation as a function of season (Figure 3a) and month (Figure 3b). While events occurred most frequently during summer time (38.4%, *p* < 0.01), the event rate was the lowest in autumn (16.4%) and winter (21.9%). Chronobiological analyses on a monthly basis identified a significant annual rhythmic pattern in TTC, which peaked in August (11.9%; *p* < 0.01) and had its nadir in November (6.3%).

## 4. Discussion

### 4.1. Baseline Characteristics

TTC is a cardiac entity which, in its clinical features, clinical course, and outcome, is not easily distinguishable from acute myocardial infarction. Because symptoms are almost the same and it is associated with a substantial risk for adverse events, first diagnosis and acute treatment remain challenging. Female patients, especially postmenopausal women, are mostly affected [1,12,13,14,15,16]. Our observations concerning the main symptoms on admission, like chest pain and dyspnoea, and the gender distribution in favor of female patients are in line with most previous reports [1,14,17,18,19]. Controversial findings regarding the underlying pathomechanism and triggering events of TTC will be objects of subsequent studies. In recent studies, excessive endogenous catecholamine expression with consequent overstimulation of the sympathoadrenergic system leading to toxic effects at the cellular level was indicated as the most likely decisive pathomechanism [20,21,22]. It is known that triggering factors, such as emotional, clinical, and physical stressors, can initiate the overstimulation of the sympathoadrenergic system. A clinical TTC onset without any stimulus is also possible. A recent multicenter analysis illustrated that not only negative emotional stressors, such as death of a beloved person or deep anger, leading to the widely known description of ‘broken heart syndrome’, but also positive life events, e.g., a grandchild’s birth, can provoke the acute heart failure syndrome called ‘happy hearts’ [17]. In series studies, admission and peak biomarkers were investigated both in TTC and in acute myocardial infarction. Cardiac ischemia levels and hsCRP of TTC patients were at most moderately increased and did not correlate with the extent of myocardial dysfunction and ECG changes compared with those of AMI patients [7,19,23,24]. TTC cases without any troponin elevation are also described in the literature [18,25]. Brain natriuretic peptide (BNP) or N-terminal-proBNP levels were much more elevated then myocardial ischemia markers [19]. Our study illustrates matching results in this context. Similarly, in line with most previous reports, the systolic left ventricular function was reduced, comprising a typical akinetic pattern and hypokinesia of the adjoining segments with markedly increased end-diastolic left ventricular pressure as a reflection of acute heart failure, in the majority of our cases [1,12,26]. Only in one case, we recorded a midventricular TTC manifestation. The observed rare right ventricular participation is in agreement with our data [27]. Meanwhile, it is suggested that physical triggers, acute neuropsychiatric disorders, high troponin levels, and a reduced left ventricular function on admission are associated with increased in-hospital complications [19].

### 4.2. Electrocardiographic Pattern

Since electrocardiographic changes of TTC patients are similar to those of patients with acute anterior STEMI, several reports analyzed the electrocardiographic pattern of both acute cardiac diseases to clarify their ECG characteristics but found limited ECG differentiation features [28,29,30]. Compared with STEMI anterior ECGs, TTC ECGs are commonly associated with more limited ST-segment elevation and pathologic Q-waves, which suggests minor myocardial damage. The overwhelming majority of the STEMI anterior patients present ST-segment elevations in derivations, reflecting the anterior (V1–V4) and, partially, the lateral wall (I, aVL), as well as reciprocal ST-changes in the inferior derivations. Lead -aVR covers apical and inferolateral regions of the heart. Diffuse ST-segment elevations in TTC, especially in lead -aVR, indicate an extensive wall motion abnormality centered around the apex, extending beyond the perfusion territory of any single coronary artery. Conversely, TTC-caused wall motion abnormalities are nontypical in the domain of V1, which mainly faces the right ventricular anterior region as well as the right paraseptal region, while ST-segment elevations in TTC were rare in lead V1. The diagnostic accuracy using the predefined electrocardiographic criteria (ST-segment elevation in -aVR with absent ST-segment elevation in V1) is identical to that reported for high-sensitivity troponin in low-risk ACS cohorts (AUC 0.79 (95%CI: 0.71–0.86)) [31]. Our results are also in agreement with those of Kosuge et al. [3]. Thus, since there is a distinct electrocardiographic pattern in TTC, what might explain this finding, especially since mostly postmenopausal women are affected? The finding might be explained by lower estrogen levels in postmenopausal women [32]. Estrogen has important regulatory effects on cardiac epinephrine metabolism and hyperpolarizes the subepicardium during ischemia and adrenergic storm [33]. Since, in the absence of estrogen, ATP-sensitive K^+^-channel density is strongly decreased, the subepicardial myocytes cannot be hyperpolarized, and the adrenergic storm is manifested by moderate ST-segment elevations. These ECG characteristics may be useful in clinical practice to distinguish TTC from AMI.

### 4.3. Seasonal Variation

Several reports indicate a seasonal variation of TTC, which has its molecular origin in the chronobiological regulation of numerous known and suspected cardiovascular risk factors for the development of TTC, which are also of decisive importance for the chronobiological distribution of AMI. The different seasonal occurrence of both diseases may be explained by a different distribution of stressful situations, which should indeed happen by chance, since in both cardiac diseases a stimulation of the sympathetic adrenergic nervous system is suggested as the main pathomechanism. Our findings concerning the chronobiological pattern of TTC are in contrast with the seasonal distribution of AMI: in the second national registry of myocardial infarction (MI), a total of 259,891 cases of AMI were analyzed, and >50% more cases were reported during winter time than during the summer [34]. Furthermore in-hospital mortality from AMI also followed a seasonal pattern and peaked in winter. Thus, as there is a distinct seasonal variation in the occurrence of TTC, what might explain this finding, especially since mostly postmenopausal women are affected? The mechanism of the disease is not yet fully understood, but excessive release of catecholamines is hypothesized to play a pivotal role in the development of TTC. In accordance, a Danish study in healthy women demonstrated a clear seasonal variation for urinary concentrations of epinephrine and norepinephrine standardized for creatinine [35]. The concentrations of urinary epinephrine were highest during the summer months, and the intra- and inter-subject variation was not explained by menstrual cycle, behavioral, emotional, or cognitive stress reactions. A further study assessed cardiovascular hemodynamics in premenopausal versus postmenopausal women (45 premenopausal and 45 postmenopausal), focusing on systemic vascular resistance (SVR) at rest and during stress [36]. After menopause, women exhibited altered sympathetic nervous system activity, and postmenopausal women were characterized by similar blood pressure but lower cardiac output and higher SVR, both at rest and during stress. In addition, postmenopausal women also had significantly higher baseline plasma norepinephrine levels and reduced cardiovascular adrenergic receptor responsiveness. Concordantly, in another study, 24 h norepinephrine excretion was significantly related both to emotional exposure and to cardiac and vascular beta-adrenergic receptor responsiveness, and data from a Japanese study suggest that reduction of estrogen levels following menopause might be involved in TTC both by indirect action on the nervous system and by direct effect on the heart [37,38]. In line with the sociodemographic distribution, recent data from a Swedish registry yielded higher suicide rates in the summer, and summertime loneliness in elderly women has been reported to be associated with TTC. Together with the ECG characteristics, the seasonal variation of TTC occurrence might be considered, when confirming the diagnosis of TTC.

## 5. Conclusions

Because the clinical features of TTC are similar to those of AMI and TTC is commonly associated with life-threatening complications, notwithstanding a completely recovery of the myocardial dysfunction, it is of decisive importance to establish criteria for the initial diagnosis and determine the appropriate treatment strategy. Therefore, it is of special interest to investigate accurate diagnostic criteria to distinguish TTC from the acute cardiac desease it mostly resembles, i.e., anterior STEMI. Our electrocardiographic analyses of the ST-segment shift both in TTC and in anterior STEMI patients indicated a simple and accurate differentiation criterion. The combined presence of ST-segment elevation in lead -aVR (e.g., ST-segment depression in aVR) and absence of ST-segment elevation in lead V1 were superior to any other electrocardiographic finding and differentiated TTC from STEMI with a sensitivity of 73%, a specificity of 84%, a positive predictive value of 63%, and a negative predictive value of 89%. The diagnostic accuracy was comparable with that of high-sensitivity (hs) troponin in a low-risk ACS group. Eventually, we suggest that the combination of the pre-defined ST changes in leads -aVR and V1 can be a useful criterion to differentiate TTC from anterior STEMI in patients who are hospitalized within 12 h of symptom onset. The results of our chronobiological analysis of TTC occurrence can also help to distinguish TTC from AMI. While TTC occurred most frequently during summer time (38.4%, *p* < 0.01), the event rate was the lowest in autumn (16.4%) and winter (21.9%). Chronobiological analyses on a monthly basis identified a significant annual rhythmic pattern in TTC, which peaked in August (11.9%; *p* < 0.01) and had its nadir in November (6.3%). AMI cases were reported mostly during winter time.

Although our data are derived from past records and more clinical characteristics of TTC have been identified in recent studies, we suggest that each result should be interpreted as a part of a totality to understand the nature of a disease, which may require a long research effort. Presumably, if our results are confirmed in large independent cohorts, they may yield diagnostic implications, changing the regular invasive AMI management in TTC patients.

## Figures and Tables

**Figure 1 biomolecules-09-00051-f001:**
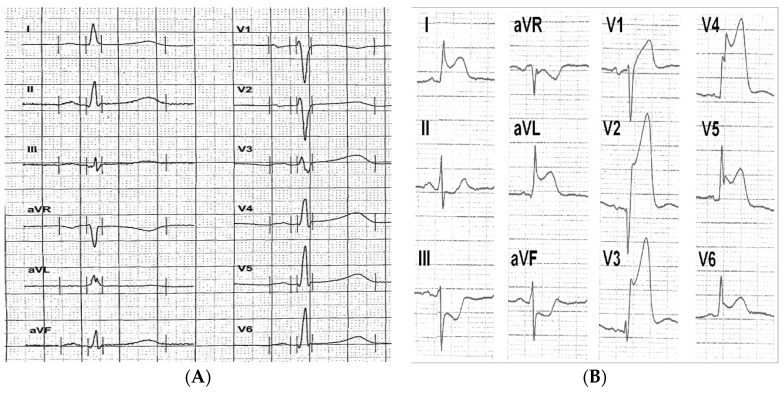
Representative electrocardiograms of (**A**) TTC and (**B**) Anterior ST-segment elevation myocardial infarction (STEMI).

**Figure 2 biomolecules-09-00051-f002:**
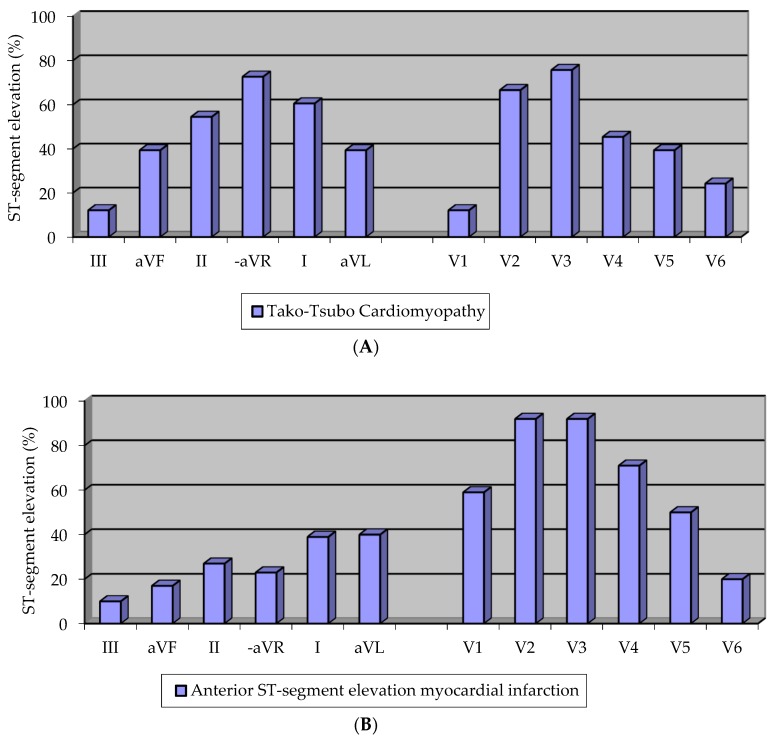
(**A**) Prevalence of ST-segment elevations in TTC; (**B**) prevalence of ST-segment elevations in anterior STEMI.

**Figure 3 biomolecules-09-00051-f003:**
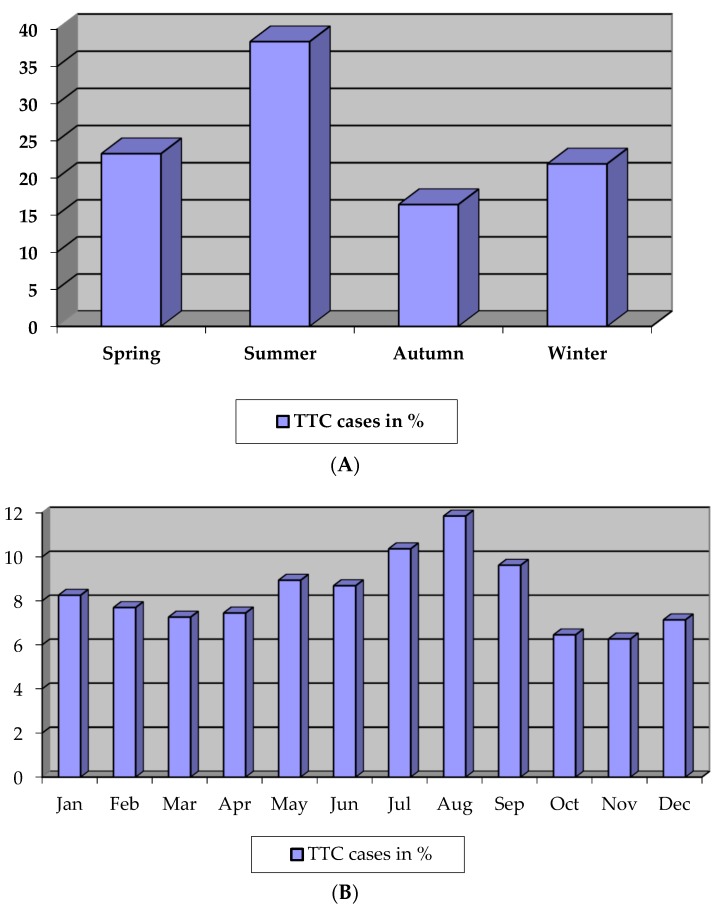
(**A**) Seasonal variation of TTC (total = 537 cases), (**B**) Monthly variation of TTC (total = 537 cases).

**Table 1 biomolecules-09-00051-t001:** Clinical characteristics of our Tako-Tsubo cardiomyopathy (TTC) patients (total number, n = 115; multicenter registered analysis)—CVE: cardiovascular event, LVEDP: left ventricular end-diastolic pressure, TIA: transient ischemic attack.

Clinical Characteristics of Patients with Tako-Tsubo Cardiomyopathy	Number of Patients (%)
Symptoms at presentation	
Chest pain	86 (74.8%)
Dyspnoea	51 (44.3%)
Palpitation	11 (9.6%)
Vertigo	7 (6.1%)
Vegetative accompanying symptoms	24 (20.9%)
Syncope	4 (3.5%)
Preceding stressors	
Emotional	53 (46.0%)
Clinical	42 (36.5%)
Physical	12 (10.4%)
Unknown trigger	8 (7.1%)
Arterial hypertension	72 (62.6%)
Hyperlipoproteinemia	48 (41.7%)
Diabetes mellitus	14 (12.2%)
Familiar disposition for CVE	26 (22.6%)
Nicotine consumption	19 (16.5%)
Chronic alcohol consumption	3 (2.6%)
Body-Mass-Index female (kg/m²)	Median: 25.5; IQR: 22.6–30.3
Body-Mass-Index male (kg/m²)	Median: 26.6; IQR: 25.0–30.3
Pre-existing obstructive coronary heart disease	0 (0%)
Pre-existing coronary artery plaque	2 (1.7%)
Reduced left ventricular ejection fraction (<55%)	72 (62.6%)
Apical left ventricular thrombus	2 (1.7%)
Increased end-diastolic left ventricular pressure	
LVEDP > 11 mmHg	68 (74.7%)
Right ventricular participation	5 (4.3%)
Obstructive Coronary artery disease (stenosis > 50%)	5 (4.3%)
In-hospital outcomes	
TIA/apoplexy	3 (2.6%)
Cardiogenic shock	5 (4.3%)
Death	3 (2.6%), all female patients
Long-term follow-up	
Death from any cause	7 (6.1% per patient-year); Median: 34 months; IQR: 13.5–63
Recurrence	10 (8.7% per patient-year); Median: 22 months; IQR: 9–33

**Table 2 biomolecules-09-00051-t002:** Clinical, emotional, and physical triggering factors of TTC in patients (total number, n = 115; multicenter registered analysis)—ALS: Amyotrophic Lateral Sclerosis.

Clinical Stressor, Number of Patients n = 42 (36.5%)	Emotional Stressor, Number of Patients n = 53 (46%)	Physical Stressor, Number of Patients n = 12 (10.4%)
Perioperative status (e.g., emergency caesarean section with atonic postpartum hemorrhage)	Death in the family	Agricultural work
Malignant disease	Family disputes	Heavy gardening
Febrile/protracted infection	Stress at work	Hiking
Acute abdomen	Excessive celebration	
Acute cholecystitis	Pronounced euphoria after	
Sepsis	grandchild’s birth	
Advanced ALS with aspiration-		
pneumonia		
Apoplexy		
Laryngitis		
Tracheomalacia		
Hypertensive crisis		
Cardiac decompensation		
Fall with rhabdomyolysis		
Vitreous hemorrhage		
Migraine		

**Table 3 biomolecules-09-00051-t003:** Admission and peak biomarkers of our TTC patients (total number, n = 115; multicenter registered analysis)—hs: high sensitivity; CK: Creatine kinase; CK-MB: Heart-type creatine kinase; NT-proBNP: N-Terminal pro-brain natriuretic peptide; hsCRP: high-sensitivity C-reactive protein.

Biomarker	Number of Patients (%)	Laboratory Value (Median; IQR)
Troponin I_on admission_	57 (49.6%)	0.42 µg/L (ng/mL); IQR: 0.11–1.52
Troponin I_maximum_	22 (19.1%)	2.84 µg/L (ng/mL); IQR: 0.9–7.65
Troponin T_on admission_	49 (42.6%)	0.2 ng/mL (µg/L); IQR: 0.09–0.66
Troponin T_maximum_	11 (9.6%)	0.43 ng/mL (µg/L); IQR: 0.16–1.12
hsTroponin_on admission_	5 (4.3%)	32 ng/L; IQR: 14–756.5
CK_on admission_	101 (87.8%)	137 U/L; IQR: 83.5–241.5
CK_maximum_	43 (37.4%)	232 U/L; IQR: 119–407
CK-MB_on admission_	83 (72.2%)	19 U/L; IQR: 7–33 U/L
CK-MB_maximum_	38 (33.0%)	18 U/L; IQR: 7.6–39.9U/L
NT-proBNP_on admission_	9 (7.8%)	2771 ng/L; IQR: 452–6147
hsCRP_on admission_	83 (72.2%)	6.6 mg/L; IQR: 2.53–36.5

**Table 4 biomolecules-09-00051-t004:** Admission biomarkers of our anterior ST-segment elevation myocardial infarction patients (total number, n = 100).

Biomarker	Number of Patients (%)	Laboratory Value (Median; IQR)
Troponin I	95 (95%)	15.8 µg/L; IQR: 2.8–81.9
CK	99 (99%)	1826 U/L; IQR: 671–3849
CK-MB	96 (96%)	165 U/L; IQR: 36.5–482.6
NT-proBNP	17 (17%)	2404 ng/L; IQR: 1346–3789
hsCRP	93 (93%)	60.9 mg/L; IQR: 13.15–167.3

**Table 5 biomolecules-09-00051-t005:** Diagnostic accuracy of ECG criteria for differentiation of TTC from STEMI of the anterior wall.

Diagnostic Accuracy	Sensitivity	Specificity	PPV	NPV
Prevalence of ST-segment elevation in -aVR (e.g., ST-segment depression in aVR) and absence of ST elevation in V1	73%	84%	63%	89%

PPV: positive predictive value; NPV: negative predictive value.

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
