# Peer review of "Electrocardiographic and Seasonal Patterns Allow Accurate Differentiation of Tako-Tsubo Cardiomyopathy from Acute Anterior Myocardial Infarction: Results of a Multicenter Study and Systematic Overview of Available Studies"

_biomolecules, 2019, doi:10.3390/biom9020051_

Round 1

Reviewer 1 Report

This study by Demir et al. represents an interesting study on Takotsubo cardiomyopathy (TTC), investigating the distinction of TTC and STEMI based on ST-depression in aVR and absence of ST-depression in V1 lead. Further, the authors have performed a multi-center study as well as a systematic analysis of the studies on seasonal variation of TTC. This paper has several interesting findings that are relevant to the clinical practice, it is quite well written, but requires critical revision of several parts:

Major comments:

·      Could the authors provide a better clinical description of the analyzed cohorts? What triggered the Takotsubo syndrome in these patietns (emotional triggers, physical triggers, etc.)? What was the ejection fraction in the compared groups? Were there other clinical parameters that would allow distinguishing TTC from STEMI?

·      Please provide a sample ECG of TTC as a figure, which would clearly demonstrate the ECG findings you found to be typical for TTC.

·      Authors defined seasonal dates different from those usually used. Why was e.g. spring considered to be from 21st of March until 20th of June?

·      Why did the authors us distinct TTC cohorts? Was this selection based on the availability of clinical data?

·      Why did the authors include studies with min. 30 TTC cases? This selection could have led to missing a number of studies.

·      It is unclear in what clinical context do the authors see the implication of their findings. The pathophysiological interrelations of catecholamines with sesonal variation is discussed, but the authors have not written why their findings are clinically relevant, except for an unclear sentence in the conclusions section. Please write a section in the discussion on the implications of your findings, and write your main findings as a conclusion.

·      What do the authors mean ‘-aVR’ lead?

Minor comments:

·      I would recommend to shorten the title. It is very long and quite confusing.

·      The abstract is structurally inaccurate, and requires rewriting. Please provide in sections, e.g. Background, objectives, methods, results, and conclusions (or in similar subheadings).

·      The initial part and several sentences in the discussion need revision of the language. The first two sentences are difficult to comprehend. Also remove repetitive sentences and phrases.

Author Response

Dear Reviewer,

thank you very much for your comments. Please find below the responses to each of your comments:

1.  We attempted a better clinical description of our TTC study population. Especially demographic  

    and clinical features, triggering factors, clinical findings (i.a.  echocardiographic/  

    ventriculographic  results existing only for TTC, laboratory values both for TTC and STEMI

    anterior- which might be a further differentiation criteria) and clinical outcome are separately  

    demonstrated. We want to mention at this point, that considering the main purpose of our       

    research  we dindn´t performed further clinical observation of our STEMI anterior cohort.

2.  We worked in representative ECGs both for TTC (own patient) and STEMI anterior (sample case)

3.  To analyze the seaonal pattern of TTC we used the seasonal definition according to the  

     astronomical calendar, which in contrast to the meteorological calendar considers  

     astrophysiological conditions (natural rotation of the earth) most likely.

4.  Since our main purpose was to analyze the  characteristic electrocardiographical     

     pattern of TTC we recruited primarily TTC-ECGs. Because we additionally intended a long-term   

     follow-up to get more clinical findings, we excluded TTC cases, in which relevant clinical data   

     were not available. Thats why we compromised our TTC cohort to a sub-cohort of 115 TTC cases  

     with available data and used these cases also for our seasonal research - there is unfortunately a  

     literal error in the description of the seasonal analyze, where the case number is reported n=  

     118, correct n=115 (total case due MEDLINE search n=537). . Because you asked for a better  

     clinical  description of our study population we used the TTC sub-cohort (n=115) in our final  

     revision.

5.   For the purpose of representativity we only included studies with at least 30 TTC patients for our  

     chronobiologigal analyze, which of course has led to missing studies.

6.   As you suggested we attempted to exemplify the implications of our findings in the discussion  

     and our main findings in the conclusion.

7.   - aVR stands for lead aVR with reversed polarity and is explained in the text (section material and  

      methods). 

8.   We shortened the title and structured the abstract

9.   We tried to provide a better revision on syntax and language. But of course if suggested, we are  

     always ready to use your English editing service.

Kind regards,

Seher Catalkaya Demir

Reviewer 2 Report

Major points

This manuscript appears to have prepared before 2011. There must be many new findings in TTC after then. You have to clarify why this paper has to be published in this moment. 

Title must be changed because it is not preferable to be written in two sentences.

The geographic characteristics play important roll in analyzing seasonal variation of a disease. You mensioned nothing about countries or regions of collecting data. You have to give us a satisfactory explanation of conducting your study like this. 

Minor points

Most of the readers including me are not familiar with the seasonal characteristics of Usak, Turkey. Please comment about it.

You described ECG with Cabrera format. I think there are many readers not used to it. Please state it clearly. 

Page 4 of 7, line 132; it must be (Figure 2) instead of (Figre 1 2).

Making figure or table is desirable for chronobiological analysis on a monthly basis.

Sentence is duplicated in conclusion part.

Author Response

Dear Reviewer ,

thank you very much for your comments. Please find below the responses to each of your comments:

 As seen on our data we conducted this research 2011 and recorded clinical variables in part retro- and in part prospectively. Of course meanwhile many new clinical findings about TTC nature are conserved. However TTC is still a multiface desease, which we are not concerned with only in the literature, but also in daily acute cardiac care practice. Since TTC is still not sufficiently understood in many aspects (i.a. pathomechanism, different seasonal occurence compared with acute myocardial infarction despite identical cardiovascular risk factors, preceding events, especially the implication of pleasant events on the neurohumoral axis and the question of available protective medication), each available statistically significant observation should be interpreted in a total setting. Eventually we suggest that our data can be also the part of further larger studies, implicating decisive diagnosis criteria. For a better understanding for our readers we explained this aspect also in the conclusion part of our manuscript: 

Although our data are derived from past period and even though increasingly more clinical characteristics about TTC are observed in recent studies, we suggest that each result should be  interpreted as a part of a totality to understand the nature of a desease, what may require a long research way throughout. Presumably, if our results can be confirmed in large independent cohorts, this may yield diagnostic implications, changing regular invasive AMI management in TTC patients.“

We shortened the title and structured the abstract.

At this point we want to mention that at the time we conducted this study about  electrocardiographic and seasonal pattern of TTC two of our authors worked in the past as MD in Southwestern Germany­­: Sibel Catalkaya, Karl-Olga Hospital, Stuttgart, Department for Cardiology until 2015 and Seher Catalkaya Demir, University Hospital of Ulm, Department for Cardiology until 2012. Therefore our study population were recruited from Southwestern Germany and are not tanged with our current working place Usak State Hospital, Turkey. 

For our seaonal analyze we additionally started a MEDLINE search and collected TTC data (methodological details are mentioned in our manuscript) derived i.a. from Europe, USA and Australia, so the geograhic impact of these regions are reflected in our results (also explained in the revision).

Please see point 3)

As you suggest the Cabrera format is explained in the manuscript.

An additional figure for demonstrating the monthly variation of TTC is worked in (please see    Fig.3b in the revision)

We tried to provide a better revision on syntax and language. But of course if suggested, we 

are always ready to use your English editing service.

Kind regards,

Seher Catalkaya Demir

Round 2

Reviewer 2 Report

I think it's incorrect to add Australian study to seasonal pattern analyzes because the weather in the southern hemisphere is quite different from that of the northern hemisphere. You should delete it.

Numberings of references are wrong and there are too many references including those not written in English.  

Author Response

Dear Reviewer,

thank you again very much for your comments.

According to your suggestion we have deleted the Australian study from our seasonal study analyze.

We also shortened our references (especially those not written in English language) and re-numbered them. 

With kind regards,

Seher Catalkaya Demir